# Strong high-energy exciton electroluminescence from the light holes of polytypic quantum dots

Xingzhi Wang[1,2,3,6], Yan Gao[4,6], Xiaonan Liu[5,6], Huaiyu Xu ®[1,2,3], Ruixiang Liu[1,2,3], Jiaojiao Song[4], Bo Li ®[1,2,3] ✉, Huaibin Shen ®[4] ✉ & Fengjia Fan ®[1,2,3] ✉

High-energy exciton emission could allow single-component multi-colour display or white light-emitting diodes. However, the thermal relaxation of high-energy excitons is much faster than the photon emission of them, making them non-emissive. Here, we report quantum dots with light hole-heavy hole splitting exhibiting strong high-energy exciton electroluminescence from high-lying light holes, opening a gate for high-performance multi-colour light sources. The high-energy electroluminescence can reach 44.5% of the band-edge heavy-hole exciton emission at an electron flux density $\Phi_e$ of $0.71 \times 10^{19}\,s^{-1}\,cm^{-2}$ – 600 times lower than the photon flux density $\Phi_p$ ($4.3 \times 10^{21}\,s^{-1}\,cm^{-2}$) required for the similar ratio. Our simulation and experimental results suggest that the oscillator strength of heavy holes reduces more than that of light holes under electric fields. We attribute this as the main reason for strong light-hole electroluminescence. We observe this phenomenon in both $Cd_xZn_{1-x}Se$-ZnS and CdSe-CdS core-shell quantum dots exhibiting large light hole-heavy hole splittings.

Typical inorganic or organic photon emitters exhibit broad absorption spectra, so the carriers can be generated by exciting the band-edge or higher-lying energy states, but those high-energy carriers thermally relax to the band-edge rapidly before they emit photons. As such, we only see a narrow band-edge emission, as generalized by Kasha in 1950[1]. Transition through phonon emission is much faster than that through photon emission, and it happens among the levels that are closely packed, because the quantized phonon energy is rather small compared with that of the photon[2–4]. In addition, the thermal occupation and oscillator strength of high-energy excitons are usually insufficient to observe strong emissions from them. Kasha's rule holds for the overwhelming majority of emitters, but researchers are

interested in finding anti-Kasha light emitters because they allow multi-colour labels and white light emission from a single emitter. Up to now, there are only a few exceptions have been found, such as organic compounds azulene[5] or thiophosgene[6], which exhibit large energy gaps between the lowest unoccupied molecular orbital (LUMO) and higher-lying energy levels.

Quantum dots (QDs) are nanocrystal semiconductors exhibiting a strong quantum confinement effect, they show much more spaced electron energy levels than that of the bulk counterparts[7–11]. However, the valence bands of II-VI and III-V semiconductors are still closely packed and the high-energy excitons are still mostly nonradiative. To the best of our literature review, higher-energy exciton emission

[1]CAS Key Laboratory of Microscale Magnetic Resonance and School of Physical Sciences, University of Science and Technology of China, Hefei, China. [2]Hefei National Laboratory, University of Science and Technology of China, Hefei, China. [3]Anhui Province Key Laboratory of Scientific Instrument Development and Application, University of Science and Technology of China, Hefei, China. [4]Key Laboratory for Special Functional Materials of Ministry of Education, National & Local Joint Engineering Research Center for High-efficiency Display and Lighting Technology, Henan University, Kaifeng, China. [5]Beijing Key Laboratory of Construction Tailorable Advanced Functional Materials and Green Applications, Experimental Center of Advanced Materials, School of Materials Science and Engineering, Beijing Institute of Technology, Beijing, China. [6]These authors contributed equally: Xingzhi Wang, Yan Gao, Xiaonan Liu. ✉e-mail: lb111@mail.ustc.edu.cn; shenhuaibin@henu.edu.cn; ffj@ustc.edu.cn

without ground state filling has been mainly observed in nanocrystals with the intermediate barrier which retards the thermal relaxation of higher-energy excitons[12].

Herein, we report that high-energy exciton electroluminescence in light-emitting diodes (LEDs) is much more viable than photoluminescence in QDs. Unlike previous observations of high-energy exciton electroluminescence that relies on filling the ground state with excessive charge carriers[13], we can achieve high-energy exciton electroluminescence at low charge concentration by increasing the relative oscillator strength. We observe that the oscillator strength of the ground-state heavy holes (HHs) can be remarkably depressed under electrical fields while that of light holes (LHs) remains intact. We demonstrate strong high-energy exciton electroluminescence in two different weakly confined QDs, both of which feature strong Stark effect and well-split LH and HH levels.

## Results

### Light hole-heavy hole splitting in polytypic quantum dots

The first type of QDs ($Cd_xZn_{1-x}Se$-ZnS based) used in this work were synthesized by modifying a protocol from literature[14,15], featuring a graded composition (Fig. 1a) and larger cores (~15.8 nm). The large core size allows us to observe a strong Stark effect in QD-LED because the wave function can be influenced greatly by the electrical field[16–18]. They were grown by utilizing the reactivity difference between CdSe and ZnSe, and an additional ultrathin ZnS shell (~0.5 nm) was grown to improve stability (Supplementary Fig. 1, see Methods for details). This radial composition change causes a composition-dependent phase transition: the Zn-rich parts tend to crystallize in a zinc-blende (ZB) structure, while the Cd-rich part tends to crystallize in a wurtzite (WZ) structure, resulting in a sandwich-like polytypic structure, as can be observed in high-resolution transmission electron microscope (HRTEM) images (Fig. 1b, c).

According to our previous density functional theory (DFT) computation[19], LH and HH degeneracy can be lifted in polytypic QDs, and the splitting between LH and HH can reach 50 meV (Supplementary Fig. 2); however, since quantum confinement of the QD used in this work (~16.8 nm in diameter) is weak, the energy levels are densely packed[8,9], resulting in overlaps of band-edge and higher energy exciton transition. Therefore, in the steady-state absorption spectrum, it is difficult to identify the absorption peaks of different energy levels.

To best distinguish the absorption peaks of LH and HH excitons, we performed femtosecond transient absorption spectroscopy, setting the pumping wavelength in resonance with the energy of the band-edge excitons. By doing so, we can only excite the band-edge electrons without generating high-energy ones. In the transient absorption spectrum of pure wurtzite cores (Fig. 2a, see Methods for synthesis details), only two transition peaks (~1.95 eV and ~2.01 eV)

can be observed, corresponding to the excitons involving the degenerate band-edge hole state (1S) and higher-order hole state (2S) respectively, as described in the literature[4,20–22]. In contrast, in our polytypic QDs, there are three transition peaks (~1.92 eV, ~1.97 eV, and ~2.03 eV), corresponding to the excitons involving the band-edge heavy-hole state ($1S_{hh}$), light-hole state ($1S_{lh}$), and higher-order hole state (2S), respectively. We also observe a decrease in the relative amplitude of the $1S_{lh}$ transition peak during another non-resonant excitation at ~2.03 eV (Fig. 2b), indicating a relaxation of LH.

To verify whether phonon bottlenecks exist between LH and HH in our QDs, we performed further femtosecond transient absorption to probe the high-energy exciton relaxation process. When we selectively pump the band-edge (pump condition 1 in Fig. 2a) and higher-lying states (pump condition 2 in Fig. 2a), we can get band-edge exciton either directly or through thermal relaxation of LH (Fig. 2c). We observed that the bleaching signal of band-edge exciton grows slower when pumping the higher-lying LH state (Fig. 2d), indicating a femtosecond time scale relaxation of LH to HH. There is a decrease in the relative change in optical density ($\Delta A$) under band-edge pumping, which is attributed to Coulomb correlation[23,24], thus we only focus on the rising edge of bleaching. To quantify the lifetime of the LH relaxation, we subtracted the rising edges of $\Delta A$ under pump conditions 1 and 2, fitted the differential with a single exponential growth dynamic (Fig. 2e), and got a lifetime of ~72 fs. These results suggest that LH and HH will reach a thermal equilibrium rapidly, without any blockage from the phonon bottleneck.

### Emission from high-energy light-hole excitons

To characterize the photoluminescence from LH and HH excitons with different exciton populations, we performed pump intensity-dependent photoluminescence spectroscopy on the polytypic QD films. The spectra can be fitted with three Gaussian peaks (Fig. 3a): a band-edge exciton photoluminescence peak ($X_1$) originating from HH excitons, and two higher-order exciton photoluminescence peaks ($X_2$ and $X_3$), originating from LH and $2S_h$ excitons respectively. The LH photoluminescence is weak at low photoexcitation intensity, which is proportional to the thermal occupation probability and oscillator strength of the LH exciton. It becomes intense when the photoexcited charge population exceeds 2 because of ground state filling. As we further increase the excitation power (photon flux density $\Phi_p$ reaches ~$1.3 \times 10^{22}$ s$^{-1}$ cm$^{-2}$), the proportion of $X_2$ photoluminescence intensity to $X_1$ photoluminescence intensity ($X_2/X_1$ intensity ratio) saturates at 45% due to nonradiative Auger recombination (see inset to Fig. 3a).

To confirm that these emission peaks originate from the different states of each single QD rather than from different QDs with different sizes, we compared the photoluminescence spectra before and after size selective gradient centrifugation (Supplementary Fig. 3), and

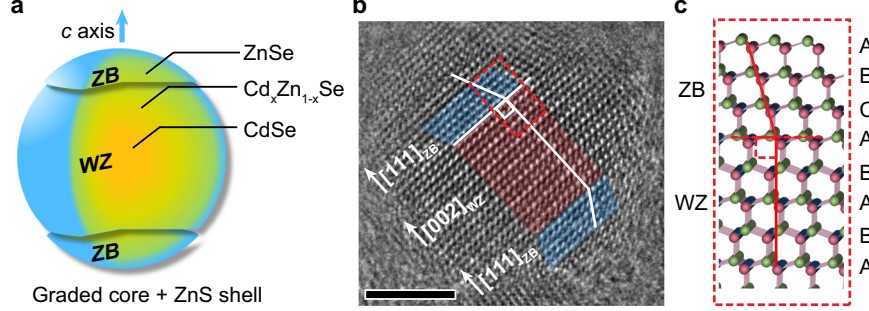

**Fig. 1 | The polytypic $Cd_xZn_{1-x}Se$-ZnS quantum dots (QDs). a** Schematic of ~15.8 nm graded $Cd_xZn_{1-x}Se$ QDs with ~0.5 nm ZnS shell. Different parts tend to differently crystallize in zinc-blende (ZB) and wurtzite (WZ) structures. **b** High-resolution transmission electron microscopy (HRTEM) image of polytypic $Cd_xZn_{1-x}Se$-ZnS QDs viewing along the [110]$_{WZ}$ and [110]$_{ZB}$ zone axes. A scale bar of 5 nm is also attached (black). **c** Schematic of ZB/WZ interface of $Cd_xZn_{1-x}Se$-ZnS QDs.

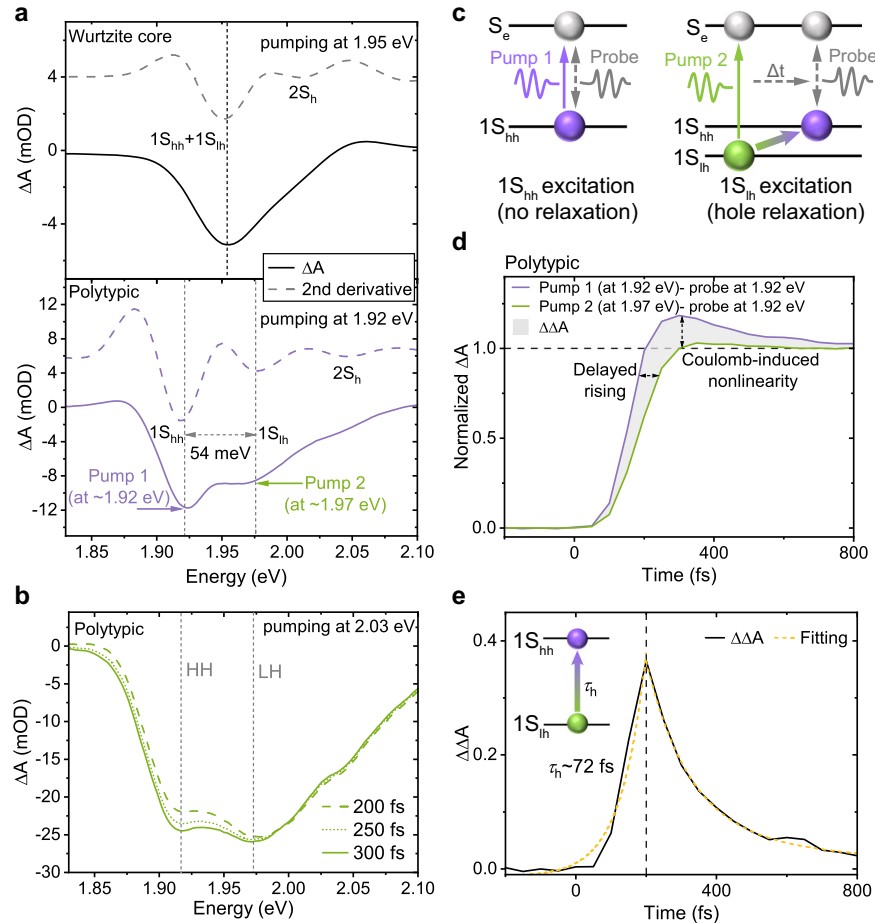

**Fig. 2 | Splitting and fast thermal relaxation between light holes (LH) and heavy holes (HH). a** Transient absorption spectra of pure wurtzite $Cd_xZn_{1-x}Se$ core (black) and polytypic $Cd_xZn_{1-x}Se$-ZnS QDs (purple) under 1.95 eV and 1.92 eV band-edge photoexcitation (at 150 fs). The second derivatives of the bleach spectra (dashed curve) allow us to quantify the splitting energy (~54 meV). The purple and green arrows indicate the photoexcitation energies. $1S_{hh}$: band-edge heavy-hole state; $1S_{lh}$: light-hole state; 2S: higher-order hole state. **b** Transient absorption spectra of polytypic $Cd_xZn_{1-x}Se$-ZnS QDs at 200 fs (dash), 250 fs (dot), and 300 fs (solid), pumping at ~2.03 eV. **c** Schematic of LH and HH photoexcitation. Under resonant HH photoexcitation (pump 1), band-edge exciton is generated immediately, while under non-resonant LH photoexcitation (pump 2), LH exciton takes some time to relax to the band-edge state ($\Delta t$). **d** Band-edge exciton generation dynamics. The difference in dynamics (shadow) is induced by both LH relaxation and Coulomb correlation, as indicated by arrows. **e** The differential dynamic of Fig. 2c. The rising edge represents the LH relaxation dynamic, which can be fitted with a single exponential function (orange, dash), and the lifetime of the light holes ($\tau_h$) can be obtained.

measured the photoluminescence spectra under both the red-edge and higher state excitation (Supplementary Fig. 4, see Methods for details). All these photoluminescence spectra exhibit similar multiple peak emissions, confirming the multiband emission originates from each single QD rather than sub-populations. Furthermore, we also observe that the lifetime of LH photoluminescence $X_2$ (~4.65 ns) is shorter than that of HH photoluminescence $X_1$ (~6.95 ns, Supplementary Fig. 5), consistent with the relatively higher oscillator strength of LH excitons.

We then measured the electroluminescence spectra of our QD-LED under different voltages (Supplementary Fig. 6; see Methods for details), which can be fitted with three Gaussian peaks as well (Fig. 3b). However, even under very low electron flux density $\Phi_e$ (~0.71 × 10$^{19}$ s$^{-1}$ cm$^{-2}$, ~1.1 A cm$^{-2}$), the proportion of LH electroluminescence is higher than that under much higher photon flux density $\Phi_p$ (~4.3 × 10$^{21}$ s$^{-1}$ cm$^{-2}$, ~2.1 kW cm$^{-2}$). When the input electron flux density exceeds 1.59 × 10$^{19}$ s$^{-1}$ cm$^{-2}$ (~2.5 A cm$^{-2}$), the $X_2/X_1$ intensity ratio can reach 80%.

It is worth mentioning that our QD-LEDs show decent luminance and efficiency with a high proportion of hot electroluminescence (Fig. 3c,d). When the $X_2/X_1$ intensity ratio approaches 57%, our device has a maximum brightness of ~80,000 cd m$^{-2}$ and an external quantum efficiency of 7.5%. Such a high proportion of high-energy exciton electroluminescence with reasonably high luminance and efficiency will potentially enable high-performance multi-colour light sources.

## Carrier concentration in quantum dot light-emitting diodes

The observed LH electroluminescence is anomalous. High-energy exciton electroluminescence has been observed in QD-LEDs under high current injection[13] (1000 A cm$^{-2}$). In our photoexcitation experiments, we also need a photon flux density of ~4.3 × 10$^{21}$ s$^{-1}$ cm$^{-2}$ (equivalent to ~9 electrons per dot) to let the proportion of LH photoluminescence reach 40%. However, under electrical excitation, we only require a very low electron flux density (~0.69 × 10$^{19}$ s$^{-1}$ cm$^{-2}$) to generate electroluminescence with a similar LH proportion.

To calculate the average electron population per QD under electrical excitation, we performed a self-developed electrically excited transient absorption (EETA) spectroscopy[25–27]. In EETA, a current pulse is used to excite the sample, and another white light laser pulse probes the change in the absorption after electrical excitation (Fig. 4a and Supplementary Fig. 7). Therefore, the electron behavior in various functional layers can be revealed, such as charge injection dynamics or equilibrated charge concentration. In our polytypic QD-LEDs, as we

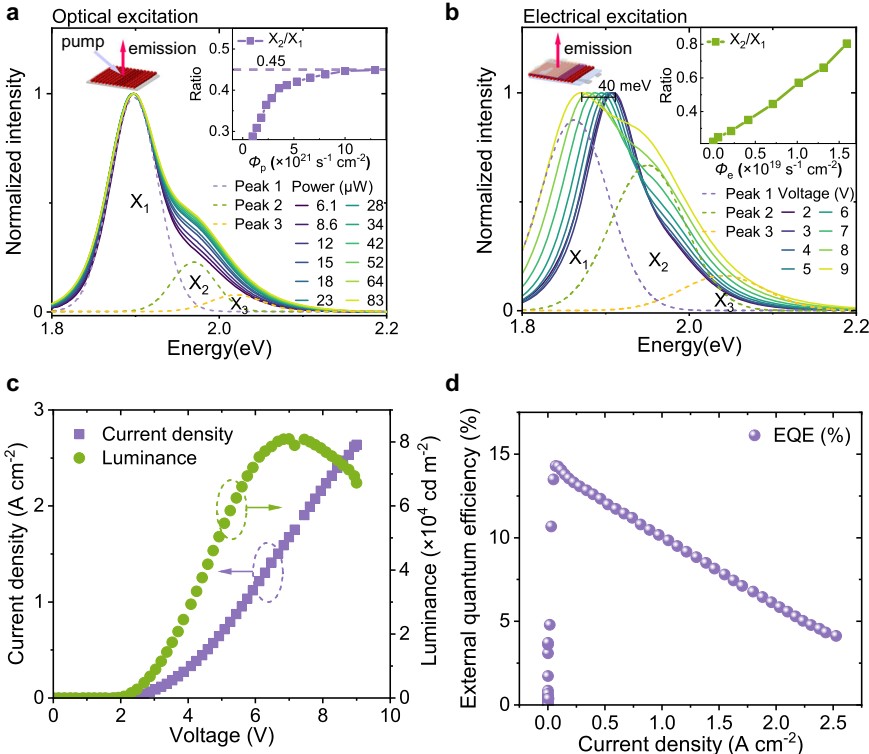

**Fig. 3 | High-energy exciton photoluminescence and electroluminescence in the polytypic Cd$_x$Zn$_{1-x}$Se-ZnS QDs. a** Pump intensity-dependent photo-luminescence spectra of the QDs film. The dashed curves represent the Gaussian fitting of three photoluminescence peaks X$_1$ (purple), X$_2$ (green), and X$_3$ (orange). Inset: the proportion of high-energy exciton emission (X$_2$/X$_1$) versus photon flux density $\Phi_p$. **b** Electroluminescence spectra under different voltages. The Stark shift of the electroluminescence peak can reach ~40 meV as we increase the driving voltage to 9 V. The dashed curves represent Gaussian fitting of three photo-luminescence peaks X$_1$ (purple), X$_2$ (green), and X$_3$ (orange). Inset: the proportion of high-energy exciton emission versus electron flux density $\Phi_e$. **c** Current density (purple squares) and luminance (green circles) of the Cd$_x$Zn$_{1-x}$Se-ZnS QD-LED as a function of voltage. **d** External quantum efficiency (EQE) of the Cd$_x$Zn$_{1-x}$Se-ZnS QD-LED as a function of injected current density.

increase the voltage from 1 V to 10 V, the amplitude of band-edge state bleaching grows continuously, indicating that electron concentration continuously increases (Fig. 4b). Comparing the amplitudes of the bleaching from EETA and nanosecond transient absorption spectra, we can estimate the average electron population in our QDs[27], which is less than 0.35 per dot under 10 V voltage (Supplementary Fig. 8). This is far less than the number of carriers required to saturate the ground state, therefore, intense LH electroluminescence observed in our experiments does not originate from ground state filling. Besides, such low injection also excludes strong heat accumulation in QD-LEDs, making the increase in thermal distribution of LH much smaller than the increase in LH electroluminescence (Supplementary Figs. 9, 10). Thus, heat accumulation cannot be the main reason for the strongly enhanced LH electroluminescence.

**Electrical-field-induced decrease in oscillator strength**
We then further investigate whether the oscillator strength of different exciton states changes under different electric fields. In our QDs, the size is large (~16.8 nm), and quantum confinement is weak, as such, electron and hole wavefunctions can be separated easier than those in QDs with stronger quantum confinement. We measured steady-state absorption spectra of the QD-LEDs under reversed biases (Fig. 4c) and found that the oscillator strength of HH is more prone to reduction than that of LH as a result of the Stark effect (Fig. 4d). We conclude here that the relative decrease in oscillator strength of the HH exciton is the main reason for the strong electroluminescence of LH (Fig. 3b).

We seek to understand this phenomenon by calculating the wavefunctions of a three-dimensional finite sphere well under electric fields[28] (Fig. 4e), with effective mass as a variable (Supplementary

Table 1, see Methods for details). The potential well deforms and the electron and hole wavefunctions spatially separate under an electric field, resulting in less wavefunction overlap (Fig. 4f), leading to a lower probability of the electron–hole recombination (i.e., a decrease in the exciton oscillator strength). We find that the wavefunction of HH will be more localized, while the wavefunction of LH tends to be more spread. Under an electric field, the wavefunction is more prone to be affected by an electrical field, i.e., easier to shift to one side of the well, resulting in more reduction of normalized wavefunction overlaps $\frac{\langle\psi_e|\psi_h\rangle_E}{\langle\psi_e|\psi_h\rangle_0}$ (~65.9% under 100 kV cm$^{-1}$) in the HH exciton oscillator strength than that of LH (~37.7% under 100 kV cm$^{-1}$). The calculation here is consistent with our experiment observation (Fig. 4c).

We also fabricated QD-LED using biaxially strained CdSe-CdS core-shell QDs[29,30], which also features weak electron confinement and well-separated LH and HH levels (Fig. 5a). These devices also exhibit strong LH electroluminescence, therefore, the hot electroluminescence mechanism we discussed is not limited to the specific chemical composition (Fig. 5b).

In summary, we report strong high-energy exciton electroluminescence from weakly quantum-confined polytypic QDs with the well split LH and HH levels. We perform optically excited transient absorption measurement and find the relaxation of LH to HH is completed within ~72 fs, excluding the phonon bottleneck as a possible origin. We also conduct electrically excited transient absorption spectroscopy measurements and observe that the maximum charge population is only 0.35 per dot, far less than the amount required to completely fill the ground state, so state filling is also excluded as a possible reason. We measure and calculate the oscillator strength of LH and HH under bias, and found that the oscillator strength of HHs

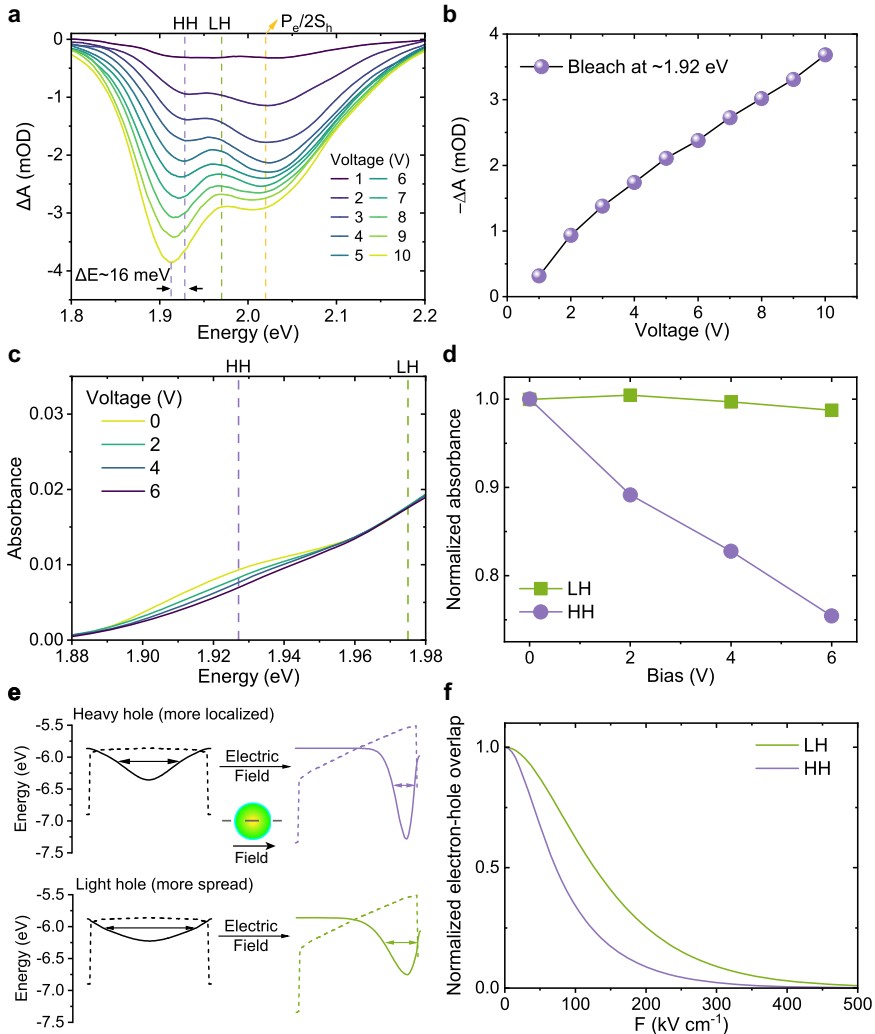

**Fig. 4 | Electric field-induced decrease of oscillator strength in $Cd_xZn_{1-x}Se$-ZnS QDs. a** Voltage-dependent steady state Electrically excited transient absorption (EETA) spectra of $Cd_xZn_{1-x}Se$-ZnS QD-LEDs, averaged within the 10th microsecond. HH: heavy hole; LH: light hole; $P_e$: higher-order electron state; $2S_h$: higher-order hole state. **b** Amplitude of band-edge state bleaching versus voltage. The increase in amplitude indicates an increasing averaged electron population $\langle N_e \rangle$. **c** Absorbance spectra under different reversed biases. **d** Normalized absorbance of LH (green, squares) and HH (purple, circles), as a function of bias. **e** The $x$-axis cross-section of HH (purple) and LH (green) wavefunctions under an $x$-direction electric field (500 kV cm$^{-1}$). The relative directions of the QD and the field are indicated by the schematic. The deformation of potential wells is represented in a dashed line. **f** Normalized wavefunction overlaps $\frac{\langle \psi_e | \psi_h \rangle_E}{\langle \psi_e | \psi_h \rangle_0}$ of LH-electron (green) and HH-electron (purple) as a function of the electric field.

reduces much more than that of LHs under electric fields, leading to enhanced electroluminescence of LHs. We observe this phenomenon in both $Cd_xZn_{1-x}Se$-ZnS and CdSe-CdS core–shell quantum dots, both of which exhibit large LH-HH splittings, indicating the finding here is not limited to the specific material composition. This work opens a gate for the development of high-performance white light sources and multi-colour light sources.

## Methods

### Materials
Cadmium oxide (CdO, 99.99%), zinc oxide (ZnO, 99.9% powder), 1-octadecene (ODE, 90%), oleic acid (OA, 90%), oleylamine (OAm, > 98% primary amine), selenium (Se, 99.99%, powder), 1-octanethiol (OT, 98%), zinc acetate dehydrate (98%) as well as various solvents were purchased from Aldrich and Aladdin.

### Preparation of the precursors
Zinc oleate ($Zn(OA)_2$): A 250 mL three-neck flask was used to mix ZnO (60 mmol), OA (60 mL), and ODE (60 mL). The mixture was then

degassed at 150 °C for 15 min. Finally, the solution was heated to 310 °C under nitrogen with vigorous stirring to get a colourless clear solution. Cadmium oleate ($Cd(OA)_2$): A 100 mL three-neck flask was used to mix CdO (10 mmol), OA (10 mL), and ODE (10 mL). The mixture was then cooled down to 150 °C to degas for 15 min. Finally, the solution was heated to 240 °C under nitrogen to obtain a colourless clear solution. Se precursor (Se-TOP): Se (10 mmol) was shot in TOP (20 mL) in a glove box under moderate heat overnight until complete dissolution. The S precursor: A 100 mL three-neck flask was used to load a desired amount of $Zn(OA)_2$ and 1-Hexanethiol (1.4 equivalent amount refers to $Zn(OA)_2$). The mixture was then degassed at 120 °C for 15 min under nitrogen with vigorous. To prevent solidification, $Cd(OA)_2$ and $Zn(OA)_2$ were kept at 70 °C and 110 °C respectively.

### Synthesis of CdZnSe /ZnS quantum dots
For the preparation of CdZnSe/ZnS core/shell QDs, a 100 mL flask was used for mixing CdO (2.4 mmol), $Zn(OAC)_2$ (4 mmol), and OA (9 mL). Then the mixture was heated to 150 °C for 30 min in flowing high-purity argon. Then, 20 mL of ODE was added to the flask, and the temperature

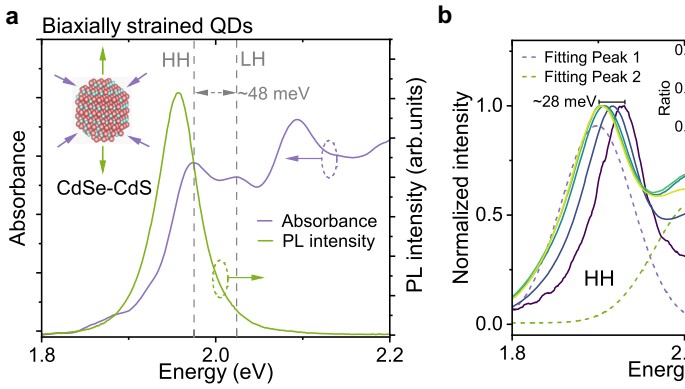

**Fig. 5 | High-energy exciton electroluminescence from biaxially strained CdSe-CdS core-shell QDs. a** Absorbance (purple) and photoluminescence spectra (green) of biaxially strained QDs. A ~ 48 meV splitting between LH and HH is highlighted (grey, dash). **b** Electroluminescence spectra of CdSe-CdS QD-LEDs under different voltages. The dashed curves represent the Gaussian fitting of photoluminescence peaks from HH excitons (purple) and LH excitons (green). Inset: the proportion of high-energy exciton emission versus voltage.

was further heated to 305 °C. 8 mL of Se precursor was injected into the reaction flask at this temperature. The reaction temperature was kept at 305 °C for 30 min for the growth of the cores. To the further growth of gradient CdZnSe cores, $Zn(OA)_2$ (4 mL) and $Cd(OA)_2$ (2 mL) were injected dropwise into the reaction solution, and Se precursor (6 mL) was added with a syringe pump (at a rate of $5\,mL\,h^{-1}$) under a temperature of 305 °C. For consecutively overcoating the ZnS shells, $Zn(OA)_2$ (5 mL) and octanethiol (2.8 mmol, 1.4 equivalent amount refers to $Zn(OA)_2$) were injected dropwise into the reaction solution with a syringe pump (at a rate of $5\,mL\,h^{-1}$). The temperature was cooled down to room temperature once the reaction was completed.

### Synthesis of biaxially strained CdSe/CdS quantum dots
For applying biaxially strain on CdSe cores, a 100 mL flask was used for mixing a hexane solution containing 200 nmol CdSe QDs with ODE (6 mL) and OAm (6 mL). The mixture was heated to 110 °C in a vacuum to evaporate the hexane, oxygen, and water. Then the reaction solution was saturated with nitrogen, and heated to 300 °C for 0.5 h. A desired amount of $Cd(OA)_2$ (diluted in 6 mL ODE) and trioctylphosphine sulfide (diluted in 6 mL ODE) solutions were injected dropwise into the as-prepared CdSe solution with a syringe pump (at a rate of $1.5\,mL\,h^{-1}$). Finally, the reaction mixture was naturally cooled to 50 °C, precipitated after 3 min centrifugation at 4323 g and then redispersed in hexane.

### Fabrication of quantum dot light-emitting diodes
For the fabrication of QD-LEDs, the patterned ITO-glass substrates were ultrasonically cleaned with detergent, deionized water, acetone, and isopropanol in sequence for 15 min. Then the substrates were dried and treated with ultraviolet ozone for 15 min. PEDOT: PSS was spin-coated on the ITO-glass substrates (4000 rpm for 40 s) and annealed at 150 °C for 15 min. Then, these substrates were transferred into a nitrogen-filled glovebox to further deposit the subsequent layers. TFB ($8\,mg\,mL^{-1}$ in chlorobenzene, 3000 rpm for 30 s), QDs ($15\,mg\,mL^{-1}$ in octane, 2000 rpm for 40 s), and ZnO ($25\,mg\,mL^{-1}$ in ethanol, 2000 rpm for 30 s) were sequentially spin-coated on the substrates under nitrogen. After spin-coating TFB and ZnO, they were annealed at 110 °C and 145 °C respectively for 30 min. Finally, a high vacuum chamber (~$5 \times 10^{-6}$ mbar) was used to thermally deposit Al cathodes (100 nm).

### Device characterization
A semiconductor parameter analyzer with a calibrated Newport silicon diode under ambient conditions was used to analyze the J–V characteristics of the QD-LEDs. The electroluminance was calibrated with a Photo Research spectroradiometer (PR735). An Ocean Optics

spectrometer (USB2000) was used to obtain the electroluminescence and time resolution electroluminescence spectra and a source meter was used for the calculations of the EQE.

### Photoluminescence measurement
A home-built confocal microscope was used to obtain the photoluminescence (PL) spectra of the film. It was excited by a high-frequency 405 nm pulse laser (Advance Laser System, PIL040X) at a 10 MHz repetition rate. The spectra were collected and analyzed by an Ocean Optics spectrometer (FLMS12313).

### Photoluminescence measurement under red-edge excitation
The solution of CdZnSe/ZnS core/shell QDs was excited by a wavelength-tunable pulse laser at a 10 MHz repetition rate. The energy of photons under red-edge excitation is controlled at ~1.85 eV (670 nm), which is on the red edge of the main peak in PL spectra (~1.91 eV) but cannot affect the higher order shoulder (~1.96 eV). Since the required power of excitation is high under red-edge excitation, a spectrum of the blank solvent under red-edge excitation is also collected to filter the signal from the excitation laser. After subtracting the spectrum of the QD solution and blank solvent, the PL spectrum of QDs under red-edge excitation is extracted and compared with the PL spectrum under higher state excitation (~2.27 eV, 545 nm). Most of the excitation laser can be filtered, but an additional peak can be observed in the red-edge excited PL of QDs because of the disturbances of the incompletely filtered excitation laser.

### Femtosecond transient absorption measurement
Transient absorption (TA) spectroscopy measurements are performed on a Time-Tech Spectra (TA100) transient absorption spectrometer. The 1030 nm femtosecond pulse (10 kHz, 190 fs) generated by a Yb: KGW laser (PHAROS, Light Conversion) is split proportionally into a probe beam (10%) and a pump beam (90%). After passing through a delay stage, the probe beam is focused on a YAG crystal to produce a white light continuum in the wavelength range of 500–1000 nm. Then the optical parametric amplifier (ORPHEUS-HE, Light Conversion) converts the pump beam into a laser pulse with a continuously tunable wavelength. The pump beam is incident on the sample at a small angle with the probe beam after passing through a chopper. The transmitted probe is collected and sent to the charge-coupled device and monochromator. The collected signal is the difference in probe intensity without and with the pump beam. According to the absorption spectrum, the probe intensity absorbed by the sample is obtained. Combining with the TA signal, the value of optical gain can be calculated

from the original probe intensity and the probe intensity when the pumping beam is present.

### Electrical excitation transient absorption measurement

An arbitrary waveform generator was used to apply the electrical pulse (1 kHz, 10 µs) with different voltages on a QD-LED sample. Meanwhile, a supercontinuum white laser (Leukos, Disco) generated a white laser pulse, which was then split proportionally into a probe beam (90%) and a reference beam (10%) by a beam splitter. An electronic time delay module controlled the delay between the probe beam sent to the excitation area and the electrical pulse applied to QD-LEDs. Then the reflected probe beam was collected by a monochromator and charge-coupled device. The reference beam was collected simultaneously to eliminate the effect of optical jitter. The collected signal was the difference in probe intensity without and with the pump electrical pulse.

### Absorption measurement under reversed bias

The white light was generated by a halogen lamp (Ideaoptics, HL2000) in the wavelength range of 360–2500 nm. Spectra reflected by the device were collected and the absorbance was calculated by an Ocean Optics spectrometer (FLMS12313). The reversed bias was applied to the device by a programmable linear DC power supply (Rigol, DP832).

### Calculation of the wavefunctions

The calculation of the wavefunctions was carried out using COMSOL Multiphysics. The models used were sphere-shaped $Cd_xZn_{1-x}Se$-ZnS core-shell QDs, and the diameters were based on our measurements. Herein the overlap between electron and hole wavefunction was represented by the product of the calculated wavefunctions under an electric field $\langle\psi_e|\psi_h\rangle_E$, and normalized to the overlap without an external electric field as $\frac{\langle\psi_e|\psi_h\rangle_E}{\langle\psi_e|\psi_h\rangle_0}$.

## Data availability

The datasets that support the findings of this study are available at Figshare (https://doi.org/10.6084/m9.figshare.26133955)[31]. More detailed data are available from the corresponding authors.

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

## Acknowledgements

We gratefully acknowledge the financial support from the National Natural Science Foundation of China (Grants No. U22A2072 [H.B.S], 62204078 [F.J.F.]), Innovation Program for Quantum Science and Technology (Grants No. 2021ZD0301603 [F.J.F.]), Fundamental Research Funds for the Central Universities (Grants No. WK3540000009 [F.J.F.], WK3540000014 [F.J.F.]), the National Key R&D

Program of China (Grant No. 2023YFE0205000 [H.B.S]), and Zhongyuan High Level Talents Special Support Plan (Grant No. 244200510009 [H.B.S]).

## Author contributions

F.J.F., H.B.S and B.L. conceived the concept and designed the experiments. X.Z.W., Y.G., and X.N.L contributed equally to this work. X.Z.W. and B.L. performed photoluminescence spectroscopy, transient absorption spectroscopy, and EETA measurement. Y.G., X.N.L., R.X.L., and J.J.S synthesized the materials, fabricated the devices, and collected the performance data of the QD-LEDs. H.Y.X. contributed to the polytypic quantum dots modeling. X.Z.W., B.L., F.J.F., and H.B.S. wrote the manuscript. All authors contributed to the scientific discussion about this work.

## Competing interests

Patents regarding the design and principle of EETA spectroscopy have been disclosed (CN Patent: CN112683797B [2021.12.14]; PCT Patent: WO/2022/121082 [2022.06.16]). B.L. and F.J.F. are the applicants and inventors of these patents. Measurement of the average electron population is performed based on this spectroscopy. The remaining authors declare no competing interests.
