## [Peer Review File · Nature Communications]

Strong high-energy exciton electroluminescence from the light holes of polytypic quantum dotsREVIEWER COMMENTS

Reviewer #1 (Remarks to the Author):

Wang et al report strong hot exciton PL and electroluminescence from the samples of core/shell quantum dots. The work has many photophysical investigations, but the key assertion is that at low intensity, the samples violate Kasha's rule. The paper has one important omission that I think should be rectified to make sense of all the remaining data: in most cases of quantum dots, shoulders or multiple emission spectra would be rationalized from the perspective of multiple populations (i.e. size). I am surprised that there is no explicit set of data to show that the two emission peaks do not derive from, for example, smaller and larger particles. Most of the remaining data can only be understood as supporting a single particle origin of the two emission colors if this explanation can be ruled out. For example, to improve this point, excitation data from various points of the emission curve should show that the sample PL derives from the same origin rather than sub-populations. Another point which makes me suspicious in this regard is the low-fluence PL of the biaxially strained samples looks very similar to literature—two PL features only occur with band filling; this element does not support the argument that two-color hot exciton emission derives from the weakly excited samples. Without directly addressing this set of controls, I would not recommend publication.

-The assertion that the light hole and heavy hole are in rapid equilibrium, is an assertion which can be supported by data. For example, pumping the heavy hole should populate the light hold level as well. Dynamics are shown, but the spectral properties are not. The relevant probe energy of the LH is higher than the data shown. The energy spacing is at least thermal energy at room temperature, or larger, so the population statistics are somewhat unclear. (Other samples appear to be closer to 90 meV in the emission spectra). The arguments of rapid relaxation from the higher energy level are inconsistent with strong hot exciton emission. For example, emission from the core of core/shell particles is very weak due to rapid transfer of holes or electrons to the core. In this case, a 72 fs transfer time would imply very limited opportunity for radiative decay.

-at one point the authors use the phrase "intense LH photoluminescence" but do not define what this means.

-The authors cite a paper on P-state emission as evidence from nanoplatelets, but in fact this work is probably not correct: the emission in nanoplatelets is more widely considered to derive from n-type trions (among other explanations). There are a few examples of bicolor emission (Moreels, Ithurria) in nanoplatelet samples with heterojunctions.

-efficacy of band filling for higher order exciton emission (no longer "hot" because it represents the population of the states) was shown by Klimov to be better under EL conditions than PL.

Reviewer #2 (Remarks to the Author):

In this work, Wang et al. report their interesting observation of light-hole hot exciton electroluminescence in two different materials systems (CdSe/ZnSe and CdSe/CdS) sharing similar LH-HH splitting and weak quantum confinement. They found the hot exciton electroluminescence is related to the effective mass-dependent Stark effect rather than state filling and phonon bottleneck effect. The manuscript is well written and the experiment results support the conclusions well. The findings provide valuable new insights into the quantum dot research field, both for fundamental understanding of physics of QDs and for

their light-emitting applications. I am happy to recommend its publication in Nature Communications if the following minor issues can be properly addressed:

1. The excitation conditions of Fig. 2b are different from those of Fig. 2a, which, however, indicated in the figure captions.
2. EETA is transient spectroscopy, but the delay times in Fig. 4a are missing. Moreover, what is reason that the amplitude of higher-lying state (at ~ 2.03 eV) bleaching is stronger than that in Fig 2?
3. According to the description in the main text, the wavefunctions are calculated based on a three-dimensional finite sphere well, but the corresponding figure (Fig. 4e) is one-dimensional. If Fig. 4e is a one-dimensional cross-section, the direction of cross section and electric field should be specified.
4. The mathematical expression and method used to calculate the wavefunction overlap in Fig. 4f should be provided in the main text and caption. The way how they normalized the curve is also missing.
5. Is the change in optical density ΔOD relative or absolute (ΔA or $\Delta A/A$)? The authors should clarify this point.

Reviewer #3 (Remarks to the Author):

The authors present an interesting perspective on the use of external fields to modulate the emission spectrum of nanocrystals. The methods are clear, very detailed and , for the electronic modulation absorption spectroscopy, also quite unique. The results that follow are intriguing and novel and merit publication in NatCom.

I do however have a concern about the mechanism, which maybe I missed something:

If one consider just a thermal distribution of holes across the split hole levels, one could think that for a splitting of ca. 50 meV, there would be roughly 10% of higher energy holes and 90% of lower energy (heavy) holes present ($\exp(-50/kbT)$) at room temperature. To boost the light hole emission over that of the heavy holes, one would then need a similar (factor of ca. 10) reduction in the heavy hole emission rate ? Can the authors develop this train of thought further with their experiments ? I think one also needs to include the natural lifetime (rate) of each level here, which due to differences in degeneracy can also nuance this.

Other than this, I think the paper presents a nice step forward in our understanding of electroluminescence in QD solids.

With kind regards.

Response Letter to *Nature Communications* submission

Paper ID: NCOMMS-23-62743

Paper Title: *Strong hot exciton electroluminescence from the light holes of polytypic quantum dots*

Dear Editor:

We sincerely thank the editor and all reviewers for their valuable feedback that has improved the quality of our manuscript. The reviewer comments are laid out below in *italicized font* and specific concerns have been numbered. Our response is given in normal font and changes/additions to the manuscript are highlighted in red.

Reviewer #1:

Comments:

Wang et al report strong hot exciton PL and electroluminescence from the samples of core/shell quantum dots. The work has many photophysical investigations, but the key assertion is that at low intensity, the samples violate Kasha's rule. The paper has one important omission that I think should be rectified to make sense of all the remaining data: in most cases of quantum dots, shoulders or multiple emission spectra would be rationalized from the perspective of multiple populations (i.e. size). I am surprised that there is no explicit set of data to show that the two emission peaks do not derive from, for example, smaller and larger particles. Most of the remaining data can only be understood as supporting a single particle origin of the two emission colors if this explanation can be ruled out. For example, to improve this point, excitation data from various points of the emission curve should show that the sample PL derives from the same origin rather than sub-populations.

Response: Thank you for your consideration. We now provide two additional sets of experiments to verify that the multiple peak emission originates from the different states of each single QD rather than from different QDs with different sizes.

We first compared the emission spectra before and after size selective gradient centrifugation. After gradient centrifugation (let those smaller QDs don't precipitate by adding insufficient anti-solvent), the average size of the QDs is larger (from 16.7 nm to 19.8 nm), and the size distribution is narrower (Figure R1a, b), resulting in narrower PL FWHM (Figure R1c), but the shoulder remains there, indicating the shoulder is not stemmed from the emission of QD with different sizes.

Figure R1. Gradient centrifugation of Cd_xZn_{1-x}Se-ZnS QDs. **a,b**, Transmission electron microscopy (TEM) image of Cd_xZn_{1-x}Se-ZnS QDs before and after gradient centrifugation. Inset: Histograms of the particle size. **c**, Photoluminescence (solid) and absorbance (dash) spectra of Cd_xZn_{1-x}Se-ZnS QDs before (red) and after (blue) gradient centrifugation. Changes in average size and size distribution of QDs result in slight differences in position and FWHM of the main peak, but the shoulder on the blue edge is almost unchanged after gradient centrifugation.

We then further compared the PL spectra measured under both the red-edge (~1.85 eV, 670 nm) and higher state excitation (~2.27 eV, 545 nm): they also match with each other (Figure R2, see Method for detail), further confirming the multiple emission originates from each single particle.

Figure R2. Normalized photoluminescence spectra of QDs under red-edge excitation (red, solid) and higher state excitation (blue, solid). High-energy photoluminescence shoulders can be observed under both red-edge and higher-state excitations. The photoluminescence spectrum is

fitted with two Gaussian peaks (blue and red, short dash), which are attributed to light-hole and heavy-hole photoluminescence, respectively. An additional peak observed on the red edge is the scattered excitation laser (grey, short dash).

2. Another point which makes me suspicious in this regard is the low-fluence PL of the biaxially strained samples looks very similar to literature—two PL features only occur with band filling; this element does not support the argument that two-color hot exciton emission derives from the weakly excited samples. Without directly addressing this set of controls, I would not recommend publication.

Response: To verify whether the multiband emission originates from electrically-driven band-filling, we adopted our recently developed electrically excited transient absorption spectroscopy (Supplementary Figure 7a), which can directly measure the electron concentration and the absorbance change (*PCT Patent WO/2022/121082, Nature Nanotechnology* **18**, 1168-1174, 2023).

Supplementary Figure 7a, Schematic of EETA spectroscopy. The QD-LED is pumped with a current pulse (1 kHz, 10 μ s), and a delayed pulsed supercontinuum white laser beam (2 kHz) is separated by a beam splitter, one of which is shined on the device, probing the change in transmission after excitation; the other is used as a reference signal to eliminate disturbances of laser intensity.

In EETA, the ground-state bleaching signal linearly grows from 1 V to 10 V voltages (Figure 4a, b), but light hole electroluminescence becomes obvious at 5 V, indicating light hole electroluminescence appears before the heavy hole state is filled. By comparing the amplitude of ground-state bleaching of EETA and conventional photoexcited nanosecond transient absorption spectroscopy (Supplementary Figure 8b), we found there are ~ 0.35 electrons per dot under 10 V. Further confirming the strong higher-order electroluminescence reported in our work is not originated from band filling.

Figure 4a, Voltage-dependent steady state EETA spectra of $\text{Cd}_x\text{Zn}_{1-x}\text{Se-ZnS}$ QD-LEDs, averaged within the 10th microsecond. **b**, The amplitude of band-edge state bleaching increases with increasing voltage, indicating an increasing averaged electron population $\langle N_e \rangle$.

Supplementary Figure 8b, Comparison of the band-edge state bleaching in power-dependent optically excited nanosecond transient absorption spectroscopy (ns-TAS, red) and voltage-dependent electrically excited transient absorption spectroscopy (EETA, blue). When a 10 V voltage is applied to the QD-LED, the bleaching of the band-edge state in EETA is equivalent to the bleaching with an averaged injected electrons of ~ 0.35 per dot in ns-TAS of the same QD-LED.

3. The assertion that the light hole and heavy hole are in rapid equilibrium, is an assertion which can be supported by data. For example, pumping the heavy hole should populate the light hole level as well. Dynamics are shown, but the spectral properties are not. The relevant probe energy of the LH is higher than the data shown. The energy spacing is at least thermal energy at room temperature, or larger, so the population statistics are somewhat unclear. (Other samples appear to be closer to 90 meV in the emission spectra). The arguments of rapid relaxation from the higher energy level are inconsistent with strong hot exciton emission. For example, emission from the core of core/shell particles is very weak due to rapid transfer of holes or electrons to the core. In this case, a 72 fs transfer time would imply very limited opportunity for radiative decay.

Response: The rapid equilibrium between the light hole and the heavy hole has been by both PL spectra under band-edge excitation (Figure R2) and transient absorption spectra (Figure 2a).

Light and heavy holes reach fast thermal equilibrium, but it doesn't conflict with emissions from both levels.

According to the Boltzmann distribution $e^{-\frac{\Delta E}{k_B T}}$, a ~ 50 meV light hole-heavy hole splitting leads to a $\sim 14\%$ proportion of light hole occupation. Furthermore, the light holes exhibit a relatively stronger oscillator strength than heavy holes (~ 1.5 times), resulting in an observable photoluminescence shoulder ($\sim 20\%$) from light holes even under low-power excitation (Supplementary Figure 2). Under an electrical field, the difference is enlarged because the oscillator strength of the heavy hole decreases at a faster pace than that of the light holes.

Based on the referee's comments, we feel the term "hot exciton" might be confusing, so we used "high-energy exciton" instead in the revised manuscript to clarify it is the emission of high-energy excitons under thermal equilibrium.

Figure 2a, Transient absorption spectra of pure wurtzite $\text{Cd}_x\text{Zn}_{1-x}\text{Se}$ core (black) and polytypic $\text{Cd}_x\text{Zn}_{1-x}\text{Se-ZnS}$ QDs (blue) under 1.95 eV and 1.92 eV band-edge photoexcitation (at 150 fs). The second derivatives of the bleach spectra (dashed curve) allow us to quantify the splitting energy (~ 54 meV). The blue and red arrows indicate the photoexcitation energies.

Supplementary Figure 2 | Excitonic structure of polytypic $\text{Cd}_x\text{Zn}_{1-x}\text{Se-ZnS}$ QDs. Left: absorbance (black) and photoluminescence (red) spectra of polytypic $\text{Cd}_x\text{Zn}_{1-x}\text{Se-ZnS}$ QDs. Different transition peaks are pointed out with arrows. The second derivative of the absorbance spectrum is also provided (violet). Right: energy levels of polytypic $\text{Cd}_x\text{Zn}_{1-x}\text{Se-ZnS}$ QDs. Transitions corresponding to the absorption peaks are noted.

4. at one point the authors use the phrase “intense LH photoluminescence” but do not define what this means.

Response: We agree with the comments. We have revised it on page 9:

“In our photoexcitation experiments, we also need a photon flux density of $\sim 4.3 \times 10^{21} \text{ s}^{-1} \text{ cm}^{-2}$ (equivalent to ~ 9 electrons per dot) to let the proportion of LH electroluminescence reaches 40%.”

5. The authors cite a paper on P-state emission as evidence from nanoplatelets, but in fact this work is probably not correct: the emission in nanoplatelets is more widely considered to derive from n-type trions (among other explanations). There are a few examples of bicolor emission (Moreels, Ithurria) in nanoplatelet samples with heterojunctions.

Response: We sincerely thank you for your correction. These suggested examples deepen our understanding. The citation and corresponding text on page 3 were revised for better expression.

“higher-energy exciton emission without ground state filling has been mainly observed in nanocrystals with the intermediate barrier which retards the thermal relaxation of higher-energy excitons.”

6. efficacy of band filling for higher order exciton emission (no longer “hot” because it represents the population of the states) was shown by Klimov to be better under EL conditions than PL.

Response: we would like to thank the referee very much for raising this question. It is a long and interesting story! This project was originally started as an electrically excited lasing project, and it has been going on for over 5 years.

When we first observed multiple EL emissions, we considered this a state-filling effect, too. We were super excited because we don’t need the high current density (up to $1,000 \text{ A/cm}^2$) and

dedicated “current focusing” architecture required in Klimov’s devices, we only need about 1 A/cm² to see the electroluminescence from higher states in conventional devices.

This observation inspired us to develop our electrically excited transient absorption spectroscopy (EETA, as introduced earlier) to measure optical gain under electrical excitation (*PCT Patent WO/2022/121082, Nature Nanotechnology* **18**, 1168-1174, 2023). Unfortunately, results reveal that the electron concentration is much lower than we thought (~0.35 per dot), nowhere near the required number for optical gain (1.4 per dot). Actually, we found in all devices we tested, electron concentration saturates at low levels (0.1-0.5 per dot) because of electron leakage, which poses a critical challenge to electrically excited lasing.

Furthermore, to observe light hole electroluminescence, we have to fill heavy holes, which requires at least 2 holes. However, this seems impossible given the hole injection is much less efficient than that of the electron. Therefore, we are quite sure the higher energy electroluminescence here doesn’t originate from the state-filling effect.

We think publishing this paper and reporting a different mechanism for electroluminescence from a higher-energy state is important because we suspect that the higher energy state emission reported in a recent paper also results from the same reason rather than state filling effect (*Advanced Materials* **34**, 9, 2022).

Figure 3b, Electroluminescence spectra under different voltages. The Stark shift of the electroluminescence peak can reach ~40 meV as we increase the driving voltage to 9 V. Inset: the proportion of higher-order exciton emission versus electron flux density Φ_e . **c**, Current density (blue squares) and luminance (red circles) of the Cd_xZn_{1-x}Se-ZnS QD-LED as a function of voltage.

Reviewer #2:

Comments:

In this work, Wang et al. report their interesting observation of light-hole hot exciton electroluminescence in two different materials systems (CdSe/ZnSe and CdSe/CdS) sharing similar LH-HH splitting and weak quantum confinement. They found the hot exciton electroluminescence is related to the effective mass-dependent Stark effect rather than state filling and phonon bottleneck effect. The manuscript is well written and the experiment results support the conclusions well. The findings provide valuable new insights into the quantum dot research field, both for fundamental understanding of physics of QDs and for their light-emitting applications. I am happy to recommend its publication in Nature Communications if the following minor issues can be properly addressed:

1. The excitation conditions of Fig. 2b are different from those of Fig. 2a, which, however,

indicated in the figure captions.

Response: We sincerely thank you for your careful consideration. The excitation conditions of Fig. 2b and Fig. 2a are at ~2.03 eV, ~1.95 eV, and ~1.92 eV respectively, which have been added in both figure and figure captions.

2. EETA is transient spectroscopy, but the delay times in Fig. 4a are missing. Moreover, what is reason that the amplitude of higher-lying state (at ~2.03 eV) bleaching is stronger than that in Fig 2?

Response: We have added the time condition in the figure caption. The signals in Fig. 4a are average values within the 10th microsecond, which will reduce the influence of injection dynamics and obtain a clear carrier distribution. The bleaching of the high-lying state at ~2.03 eV will be stronger because the injection of electrons with higher energy will increase when the driving voltage increases. These electrically excited high-lying electrons will keep a longer lifetime than those photoexcited high-lying electrons, because electrons and holes are separately injected into the device, and high-lying electrons are hard to relax without holes.

3. According to the description in the main text, the wavefunctions are calculated based on a three-dimensional finite sphere well, but the corresponding figure (Fig. 4e) is one-dimensional. If Fig. 4e is a one-dimensional cross-section, the direction of cross section and electric field should be specified.

Response: Fig. 4e is a cross-section of the three-dimensional finite sphere well. The direction of the cross-section has been added in the figure caption and the direction of the electric field has been added in Fig. 4e.

4. The mathematical expression and method used to calculate the wavefunction overlap in Fig. 4f should be provided in the main text and caption. The way how they normalized the curve is also missing.

Response: We have added the formula $\frac{\langle \psi_e | \psi_h \rangle_{\vec{E}}}{\langle \psi_e | \psi_h \rangle_0}$ in the main text (page 10) and the figure caption.

Details about the calculation of the overlap have also been added to the methods:

“Herein the overlap between electron and hole wavefunction is represented by the product of the calculated wavefunctions under an electric field $\langle \psi_e | \psi_h \rangle_{\vec{E}}$, and normalized to the overlap without an external electric field

$$\frac{\langle \psi_e | \psi_h \rangle_{\vec{E}}}{\langle \psi_e | \psi_h \rangle_0} ,,$$

5. Is the change in optical density ΔOD relative or absolute (ΔA or $\Delta A/A$)? The authors should clarify this point.

Response: The change in optical density ΔOD is a relative change ($\Delta A/A$). The corresponding description on page 5 has been revised. Thank you for your careful observation.

Reviewer #3:

Comments:

The authors present an interesting perspective on the use of external fields to modulate the

emission spectrum of nanocrystals. The methods are clear, very detailed and , for the electronic modulation absorption spectroscopy, also quite unique. The results that follow are intriguing and novel and merit publication in NatCom.

I do however have a concern about the mechanism, which maybe I missed something:

If one consider just a thermal distribution of holes across the split hole levels, one could think that for a splitting of ca. 50 meV, there would be roughly 10% of higher energy holes and 90% of lower energy (heavy) holes present ($\exp(-50/kbT)$) at room temperature. To boost the light hole emission over that of the heavy holes, one would then need a similar (factor of ca. 10) reduction in the heavy hole emission rate ? Can the authors develop this train of thought further with their experiments ?

Response: According to the Boltzmann distribution $e^{-\frac{\Delta E}{k_b T}}$, the population of light holes should be about 14% of heavy holes in this work. Besides, the oscillator strength (i.e., emission rate) of light holes is about 1.5 times higher than that of the heavy holes, according to the absorption spectra (Supplementary Figure 2). As a result, the emission from light holes should be about 20% of the emission from heavy holes, which is consistent with the photoluminescence spectra.

To let light hole emission reach 40% of the heavy hole, we only need a 2-fold increase in the relative emission rate of the light hole, which is easy to achieve as we have shown in this work.

However, to obtain light hole emission stronger than that of the heavy holes, we need high driving voltage, which is interesting but challenging in the experiment because of the severe efficiency roll-off and damage in QD-LEDs. When we solve this restriction, the utilization of the multi-band emission will be more valuable.

Supplementary Figure 2 | Excitonic structure of polytypic $Cd_xZn_{1-x}Se-ZnS$ QDs. Left: absorbance (black) and photoluminescence (red) spectra of polytypic $Cd_xZn_{1-x}Se-ZnS$ QDs. Different transition peaks are pointed out with arrows. The second derivative of the absorbance spectrum is also provided (violet). Right: energy levels of polytypic $Cd_xZn_{1-x}Se-ZnS$ QDs. Transitions corresponding to the absorption peaks are noted.

I think one also needs to include the natural lifetime (rate) of each level here, which due to differences in degeneracy can also nuance this.

Response: The photoluminescence decays of light-hole emission and heavy-hole emission are

added as Supplementary Figure 5. We used a monochromator to separate these two bands and observed that the lifetime of light-hole emission is shorter than that of heavy-hole emission, which is consistent with the result that the oscillator strength (i.e., emission rate) of light-hole is higher than heavy-hole, as observed in the absorption spectrum.

Supplementary Figure 5 | Time-resolved photoluminescence intensities of LH (red) and HH (blue) excitons in $\text{Cd}_x\text{Zn}_{1-x}\text{Se}$ QDs in log coordinate, fitted with single exponential decay. The lifetimes of LH and HH photoluminescence are given respectively.

REVIEWERS' COMMENTS

Reviewer #1 (Remarks to the Author):

I support publication as reviewer comments have been answered reasonably in the revised work.

Reviewer #2 (Remarks to the Author):

The authors have done an excellent job in preparation the responses to my and other reviewers' comments. The paper should be published in Nat Commun as is.

Reviewer #3 (Remarks to the Author):

To whom it may concern,

The Authors have presented detailed and clear explanations answering my small remaining concerns and I think the manuscript is, in my view, suitable for final publication. The work is clearly novel and of great interest to those working to optimize and understand electroluminescence from colloidal nanocrystals, a field that requires further development to date.

best regards.